# Inhibitory Effects of Erythrosine/Curcumin Derivatives/Nano-Titanium Dioxide-Mediated Photodynamic Therapy on *Candida albicans*

**DOI:** 10.3390/molecules26092405

**Published:** 2021-04-21

**Authors:** Kasama Kanpittaya, Aroon Teerakapong, Noppawan Phumala Morales, Doosadee Hormdee, Aroonsri Priprem, Wilawan Weera-archakul, Teerasak Damrongrungruang

**Affiliations:** 1Division of Periodontology, Department of Oral Biomedical Sciences, Faculty of Dentistry, Khon Kaen University, Khon Kaen 40002, Thailand; kanpitaya@kkumail.com (K.K.); arotee@kku.ac.th (A.T.); nootdoosadee@hotmail.com (D.H.); 2Dental Department, Chumphae Hospital, Khon Kaen 40130, Thailand; 3Laser in Dentistry Research Group, Khon Kaen University, Khon Kaen 40002, Thailand; 4Department of Pharmacology, Faculty of Sciences, Mahidol University, Bangkok 10400, Thailand; noppawan.phu@mahidol.ac.th; 5Faculty of Pharmacy, Mahasarakham University, Maha Sarakham 44150, Thailand; aroonsri@kku.ac.th; 6Division of Dental Public Health, Department of Preventive Dentistry, Faculty of Dentistry, Khon Kaen University, Khon Kaen 40002, Thailand; wilwee@kku.ac.th; 7Research and Academic Services, Khon Kaen University, Khon Kaen 40002, Thailand; 8Division of Oral Diagnosis, Department of Oral Biomedical Sciences, Faculty of Dentistry, Khon Kaen University, Khon Kaen 40002, Thailand

**Keywords:** biofilm, erythrosine, curcumin derivatives, *Candida albicans*, photodynamic therapy

## Abstract

This study focuses on the role of photosensitizers in photodynamic therapy. The photosensitizers were prepared in combinations of 110/220 µM erythrosine and/or 10/20 µM demethoxy/bisdemethoxy curcumin with/without 10% (*w/w*) nano-titanium dioxide. Irradiation was performed with a dental blue light in the 395–480 nm wavelength range, with a power density of 3200 mW/cm^2^ and yield of 72 J/cm^2^. The production of ROS and hydroxyl radical was investigated using an electron paramagnetic resonance spectrometer for each individual photosensitizer or in photosensitizer combinations. Subsequently, a PrestoBlue^®^ toxicity test of the gingival fibroblast cells was performed at 6 and 24 h on the eight highest ROS-generating photosensitizers containing curcumin derivatives and erythrosine 220 µM. Finally, the antifungal ability of 22 test photosensitizers, *Candida albicans* (ATCC 10231), were cultured in biofilm form at 37 °C for 48 h, then the colonies were counted in colony-forming units (CFU/mL) via the drop plate technique, and then the log reduction was calculated. The results showed that at 48 h the test photosensitizers could simultaneously produce both ROS types. All test photosensitizers demonstrated no toxicity on the fibroblast cells. In total, 18 test photosensitizers were able to inhibit *Candida albicans* similarly to nystatin. Conclusively, 20 µM bisdemethoxy curcumin + 220 µM erythrosine + 10% (*w/w*) nano-titanium dioxide exerted the highest inhibitory effect on *Candida albicans*.

## 1. Introduction

*Candida albicans* is one of the most common fungal infections in humans, acting as normal flora and an opportunistic pathogen, which colonizes at several anatomically distinct sites, including the skin, mouth, gastrointestinal tract, and vagina. It is found that in healthy people, approximately 40% of this *Candida* species is detected in the saliva and oral mucosa. It can cause severe diseases, from deep mucous membrane infections to systemic infections [1]. Systemic candidiasis infections may be fatal, such as candidemia, which has a mortality rate of 35% [2]. The infection is frequently found in patients with immunocompromised disorders, such as HIV.

In the oral cavity, *C. albicans* can be found on the palate, vestibular buccal mucosa, tongue, sublingual tissue, and saliva. This species is rarely found in the gingival sulcus of healthy individuals [3]. However, an increase in the number of these infections in oral biofilms, including the gingival sulcus, is associated with the occurrence and development of periodontitis [4]. There is a 7.1–19.6% possibility of *Candida albicans* being found in patients with chronic periodontitis [5]. In individuals with cancer, diabetes, and immunodeficiency diseases such as HIV or AIDS, who will have a higher prevalence of *Candida albicans*, this species will cause the rapid progression of periodontitis.

Currently, nystatin oral suspension, an antifungal that suppresses fungal germination, is used as a “first-line” agent against intraoral candidiasis infection. This antifungal has some drawbacks; for instance, one must use the medication for a long time, many times per day, and it may cause drug resistance. There are also some side effects, such as abdominal distress, diarrhea, nausea, vomiting, dysgeusia, loss of appetite, and intra-oral irritation. In more severe cases, rashes or hives, arrhythmia, bronchospasm, dyspnea, and facial swelling may occur. Therefore, developing a novel treatment modality with minimal adverse effects, such as photodynamic therapy, is crucial to solving the above mentioned adverse conditions.

Photodynamic therapy (PDT) is a process that uses non-toxic substances known as photosensitizers together with low-intensity visible light in an intra/intercellular oxygenated environment to stimulate the formation of free radicals, for instance, reactive oxygen species (ROS) [6]. Ideal photosensitizers should specifically penetrate only into target cells or tissues, and not the adjacent normal cells/tissues. In addition, the irradiation can be restricted to an area that has a relatively high concentration of photosensitizers, making photodynamic therapy useful in spatial selectivity. This promising method can be used for a variety of applications, such as microbial inhibition, and the treatment of infectious diseases [7], tumors, cancer [8], and certain skin diseases [9]. To our knowledge, there are almost no reports regarding the microorganism’s resistance to photodynamic therapy. However, it can kill many pathogenic microorganisms that are resistant to drugs and disinfectants [7].

The photosensitizers used in this study were erythrosine and curcumin derivatives. Erythrosine has the ability to absorb light across a wide range of wavelengths, making it suitable to be used as a photosensitizer in the photodynamic treatment process with multiple light colors, such as blue light and green light. In 2012, Costa et al. [10] studied the killing of *Candida albicans* and *Candida dubliniensis* using a 532 nm green laser with an energy density of 42.63 mW/cm^2^. It was found that erythrosine at concentrations above 3.12 µM was able to kill planktonic *Candida albicans*, while the fungicidal effects of the *Candida albicans* biofilm require concentrations above 400 µM. In the same year, Costa et al. [11] studied Rose bengal, one of the more potent phenothiazine stains, and found it could significantly reduce *Candida albicans* numbers in planktonic and biofilm form when used at 10 and 40 µg/mL, respectively, and irradiated with a blue laser at 455 ± 20 nm with an energy density of 95 J/cm^2^. Despite the significant reduction in *C. albicans* numbers, this study achieved less than a 1log_10_ reduction in *Candida albicans* in biofilm; with this level of reduction, it is not effective in a clinical situation, and thus more effective means are required.

Interestingly, curcuminoid, the most extensively investigated phytochemical with a wide range of therapeutic properties, which include antimicrobial and anticancer, is also emerging as a photosensitizer for photodynamic therapy [12]. Three types of curcuminoid derivatives are found in nature: curcumin, demethoxycurcumin, and bisdemethoxycurcumin. The presence of the methoxy groups in the structure reduces the disinfection ability of this substance when used in the process of photodynamic treatment. According to Tønnesen et al., in 1995, while studying the properties of curcumin and its various derivatives [13], it was found that the presence of the methoxy group next to the phenolic group makes the hydroxyl group in curcumin a stronger hydrogen bond receptor than the hydroxyl groups in demethoxycurcumin and bisdemethoxycurcumin, thus both derivatives are more likely to be a hydrogen bond donor. Therefore, curcumin tends to interact with sites possessing hydrogen bond donor properties, while demethoxycurcumin and bisdemethoxycurcumin are very attractive hydrogen bond acceptor binding sites. This gives bisdemethoxycurcumin a superior ability to attach to bacterial cell walls through hydrogen bonds and cause bacterial phototoxicity [14]. The cell wall of the fungus is chitin, a substance that is a carbohydrate and has a structure similar to peptidoglycans in the bacterial cell wall, so it is hypothesized to exert a similar result when tested in fungus. In addition, in 2006, Jayaprakasha et al. studied the antioxidant activities of all three curcumin derivatives using the linoleic acid peroxidation method. It was found that curcumin has the strongest antioxidant properties, followed by demethoxycurcumin and bisdemethoxycurcumin, respectively. It is possible that when the free radicals are released, curcumin inhibits the free radicals on its own, due to its antioxidant properties [15]. Curcumin has an absorption peak range of 300–500 nm, with a maximum absorption band at wavelength 430 nm [16], which corresponds to the blue light range. Therefore, the present study chose to use a resin composite curing unit as a light source, because it is already available in every dental clinic worldwide.

The free radicals that occur in the photodynamic treatment process are derived from two types of reactions, depending on the type of free radicals that occurs in the reaction. Type I is caused by the transmission of electrons between the photosensitive stimulant in the form of triplets and many other molecules, such as the free radicals of hydroxyl radical, superoxide, peroxide. The type II reaction, which is mediated by energy transfer, will mainly release singlet oxygen [17]. The products from the type I reaction exert more anti-microorganisms than the type II reaction because the hydroxyl radical is more toxic than singlet oxygen. These are the most active free radicals, causing the most harm among the free radicals [18], and are also the most sensitive free radicals [19] that are difficult to eliminate compared with singlet oxygen [20].

Titanium dioxide is a substance that is often used as a pigment in light distribution because it can absorb light in almost all spectra, with three main properties: reflection of light from the surface of the crystal; refraction of light inside the crystal, and diffraction of light. Therefore, it is mixed with photosensitive substances in order to reflect and refract the light. Nano-titanium dioxide is titanium dioxide in nano-size, and is one of the most widely used nanomaterials in the world. It has a better catalytic capability compared with the larger-sized titanium dioxide [21]. Nano-titanium dioxide has three basic structures: anatase, rutile, and brookite. Anatase is the most stable form [22]. The study of Teerakapong et al. compared the effects of photodynamic therapy using erythrosine and blue light with/without 1% *w/w* nano-titanium dioxide (anatase form). It was found that the addition of anatase nano-titanium dioxide could significantly increase the antifungal effects against *C. albicans* in biofilm form [23].

Currently, no single photosensitizer can stimulate the reaction of both types of free radicals at the same time. Thus, we aimed to use erythrosine mixed with curcumin derivatives in the presence or absence of nano-titanium dioxide in order to concurrently give rise to both types of photodynamic reactive oxygen species products. We then elucidated each of the above regimens that are capable of exerting antifungal effects in biofilm forms in an in vitro setting. Additionally, we expected that this photosensitizer would not be toxic to human cells; therefore, testing the cytotoxicity with a human gingival fibroblast cell line via PrestoBlue^®^ was also to be undertaken.

## 2. Results

### 2.1. ROS Determination Using Electron Paramagnetic Resonance Spectrometer

The total ROS production was quantified with TEMPO, and is shown in Figure 1. The combination of 20 µM demethoxycurcumin and 220 µM erythrosine with 10% nano-titanium dioxide was the preparation that produced the highest ROS, 56.13 ± 4.12 µM, whereas 20 µM bisdemethoxycurcumin produced the lowest ROS, at 16.66 ± 3.61 µM. When compared among the various combinations of utilized photosensitizers, it was found that the combination of three photosensitizers enabled significantly greater ROS quantification than single photosensitizers. 

The comparison between curcumin derivatives at the same concentration levels is illustrated in Figure 1a. The combination with bisdemethoxycurcumin was likely to produce more ROS than the combination with demethoxycurcumin, but there were no statistically significant differences except the following: the combination of 10 µM bisdemethoxycurcumin with 110 µM erythrosine > the combination of 10 µM demethoxycurcumin with 110 µM erythrosine (*p* = 0.019); the combination of 10 µM bisdemethoxycurcumin with 220 µM erythrosine > the combination of 10 µM demethoxycurcumin with 220 µM erythrosine (*p* < 0.001), and the combination of 20 µM bisdemethoxycurcumin with 220 µM erythrosine > the 20 µM combination of demethoxycurcumin with 220 µM erythrosine (*p* = 0.015).

Figure 1b demonstrates that the combination of 10 µM demethoxycurcumin with erythrosine with 10% nano-titanium dioxide significantly increased ROS production, accounting for 1.52–2.36-fold (*p* < 0.001) more than 10 µM demethoxycurcumin alone. The combination with a low concentration of erythrosine without titanium dioxide did not show an enhanced effect over 10 µM demethoxycurcumin alone. 

For higher concentrations of demethoxycurcumin, the combination with other compounds produced higher levels of ROS than demethoxycurcumin alone (1.42- to 2.52-fold (*p* < 0.05)), except for the double combination with nano-titanium dioxide (Figure 1c).

Regarding bisdemethoxycurcumin, all double and triple combinations generate ROS at higher levels than the single photosensitizer, ranging from 1.35-fold to 2.02-fold (*p* < 0.05) (Figure 1d,e).

Figure 2 compares the total ROS produced between demethoxy curcumin and bisdemethoxy curcumin at the same concentrations. It is clear that most bisdemethoxycurcumin tended to give higher total ROS levels (1.0-fold to 1.4-fold) and this reached the significant level only with the concentration of 10 µM with 110 or 220 µM of erythrosine, and with the concentration of 20 µM with 220 µM of erythrosine, at *p* = 0.019, *p* < 0.001 and *p* = 0.015, respectively.

The effects of nano-titanium dioxide on ROS formation were analyzed (Figure 3). The results showed that in the demethoxycurcumin and bisdemethoxycurcumin groups, adding 10% nano-titanium dioxide by mass did not contribute to the formation of ROS. However, in the erythrosine groups and other erythrosine-containing test reagents, the addition of 10% nano-titanium dioxide by mass resulted in a significantly higher amount of ROS (*p* < 0.05). The ROS produced from erythrosine at concentrations of 110 and 220 µM were 22.7 ± 0.61 µM and 30.08 ± 1.78, respectively. When we added up 10% nano-titanium dioxide by mass, the ROS produced were at the levels of 32.27 ± 2.45 µM and 35.42 ± 0.62, which were increases of 1.4-fold and 1.2-fold, respectively. The addition of 10% nano-titanium dioxide (by mass) induced the maximum increase in ROS produced from the combination of 20 µM demethoxycurcumin and 220 µM erythrosine, from 36.94 ± 2.13 to 56.13 ± 4.12, or a 1.5-fold increase (*p* < 0.0001). Interestingly, in the bisdemethoxycurcumin group, the addition of 10% nano-titanium dioxide (by mass) caused less escalation in the accumulation of ROS (17–35% increase).

Type I ROS production was investigated via the spin trapping technique using DMPO. All of the photosensitizer preparations with curcumin derivatives immediately facilitated the generation of hydroxyl and superoxide anion radicals after irradiation with blue light. The ESR signals of the DMPO-OH and DMPO-OOH adducts at 30 s after irradiation are shown in Figure 4. We observed that the ESR signal of the DMPO-OH adduct rapidly decayed; therefore, the signal of DMPO-OOH could indicate the superoxide anion radicals and the conversion of DMPO-OH adduct.

The ESR signal intensities of the DMPO-OH adduct produced from 10 µM demethoxycurcumin or bisdemethoxycurcumin alone were 136 ± 12 and 129 ± 20 A.U. The ESR signal tended to increase in the combination with erythrosine or/and 10% nano-titanium dioxide; however, statistical significance was not observed. It should be noted that ESR signals were not observed in conditions without curcumin derivatives.

### 2.2. Viability of Gingival Fibroblast Cells

The testing of the viability of the test photosensitizers on gingival fibroblast cells was performed via PrestoBlue^®^. As shown in Figure 5, at 6 h, the percentage of gingival fibroblast cell viability was between 75.98 ± 5.28 and 95 ± 7.72%. Additionally, at 24 h, the percentage of gingival fibroblast cell viability was 74.73 ± 10.70–98.02 ± 23.66%.

When compared with the fibroblast cells cultured in DMEM (positive control), the percentages of gingival fibroblast cell viability showed statistically significant differences in all samples (*p* < 0.05), except in 10 μM demethoxycurcumin + 220 μM erythrosine, 10 μM bisdemethoxycurcumin + 220 μM erythrosine, and 20 μM bisdemethoxycurcumin + 220 μM erythrosine at 24 h. When compared with 50% (*w*/*w*) hydrogen peroxide (negative control), the viable cell number for all test photosensitizers in all periods was significantly higher than for hydrogen peroxide (*p* < 0.05).

No significant difference was observed in terms of cell numbers between the test photosensitizers with and without nano-titanium dioxide.

### 2.3. Candida Albicans Inhibition

Figure 6 shows that all tested photosensitizers reduced the *Candida albicans* biofilm significantly compared to phosphate buffer saline (*p* < 0.05). When compared with oral nystatin suspension, 18 test photosensitizers displayed a similar ability to reduce the *Candida albicans* biofilm to that shown by the oral nystatin suspension. The greatest reductions were achieved with 20 µM bisdemethoxycurcumin + 110 µM erythrosine and 20 µM bisdemethoxycurcumin + 110 µM erythrosine + 10% nano-titanium dioxide (by mass), with approximately 1.1log_10_ CFUs/mL compared to PBS. The lowest reduction was achieved with 110 µM erythrosine + 10%Nano-titanium dioxide (by mass), with only 0.4 log_10_ CFUs/mL from PBS. It was noted that bisdemethoxycurcumin tended to inhibit *Candida albicans* more effectively than demethoxycurcumin. Interestingly, 10 µM bisdemethoxycurcumin + 110 µM erythrosine + 10% nano-titanium dioxide (by mass) had a significantly greater anticandidal effect than 10 µM demethoxycurcumin + 110 µM erythrosine + 10% nano-titanium dioxide (*p* < 0.001). Nano-titanium dioxide did not affect *C. albicans* suppression among the groups tested.

### 2.4. The Correlation of ROS Production and Candida albicans Inhibition

According to the analysis of the Spearman’s rank correlation coefficient shown in Figure 7, the number of ROS produced, and their antifungal effects, were significant and inversely related to each other (*p* < 0.05). The higher the reactive oxygen species content, the lower the fungal number, with a relative coefficient (r) of −0.267, or low–moderate.

## 3. Discussion

Our study demonstrated successful *C. albicans* biofilm inhibition, mediated by a large amount of ROS formation, by curcuminoid and erythrosine in combination with nano-titanium dioxide. Our photosensitizer regimen was not cytotoxic to normal human gingival fibroblast cells for up to 24 h.

In photodynamic therapy, multiple colors and wavelengths of light can be used, but the reason for using blue light in this experiment was that the trial aimed to develop a topical oral antifungal treatment. Blue light is a light that is available in dental clinics and is used to polymerize composite resin for tooth-like fillings. Moreover, the main photosensitizer used in the present study has a maximum absorption ability in the blue light region [16]. Therefore, it is suitable for application in oral antimicrobial photodynamic therapy.

### 3.1. Candida albicans Inhibition

The results obtained from the present study show that the greatest *C. albicans* reductions occurred when induced by 20 µM bisdemethoxycurcumin + 110 µM erythrosine and 20 µM bisdemethoxycurcumin + 110 µM erythrosine + 10% nano-titanium dioxide. The aforementioned photosensitizers could reduce *Candida albicans* by 1.12log_10_ compared to the negative control (PBS). These reductions were less than the 1.4log_10_ reduction achieved by Teerakapong et al. using 220 µM erythrosine + 1% nano-titanium dioxide, irradiated with dental blue light at a wavelength of 395–480 nm and power density of 1200 mW/cm^2^ for 60 s, yielding the same energy density at 72 J/cm^2^ but in a continuous mode. This inconsistency was partly due to the difference in light system and irradiation mode; our study used the pulse mode with irradiation for 3 s and a 1 s rest between each session. The study of Miyamoto, Umebayashi, and Nishisaka [25] reported that the cytotoxic effect of the pulsed mode of irradiation was lower than that of the continuous light system. Additionally, a study by Klimenko [26] demonstrated that the higher the power density, the lower the singlet oxygen formation. In the present study, we utilized a power density that was 3200 mW/cm^2^ higher than that of Teerakapong et al., and thus the ROS formed should be at a low level. The irradiation time was another concern; normally, antimicrobial PDT needs 60 s or more [10,12,23], but the present study used only 21 s of irradiation, which was probably inadequate for the completion of the photodynamic reaction, hence the incomplete *C. albicans* inhibition. 

### 3.2. ROS in Photodynamic Treatment

In the present study, it was found that curcumin derivatives exerted the main antifungal effect. In many samples, the fungicidal effects of curcuminoids masked the antifungal effects of erythrosine. This is expected to be caused by the difference in PDT product yield. Curcumin derivatives are able to produce type I ROS (mainly hydroxyl radicals), while erythrosine produces type II ROS (singlet oxygen). Generally, the hydroxyl radical is more toxic than singlet oxygen, because hydroxyl radicals are the most active, the most hazardous [18], the most sensitive [19], and the most difficult to eliminate among the ROS [19]. To the best of our knowledge, our study is one of the first studies to discover that curcumin derivatives not only yielded the hydroxyl radical alone, but also produced high levels of superoxide anion, especially during the first 10 s after light exposure. Superoxide was the ROS belonging to type I to be derived in the photodynamic reaction. This ROS product was classified as a moderate reactor in electron reduction [27]. Altogether, the type I ROS product derived from curcuminoid could synergistically provoke an antifungal effect.

In addition, Jayaprakasha et al. studied the antioxidant ability of the three curcumin derivatives using the linoleic acid peroxidation method. Their study found that curcumin possesses the highest antioxidant properties, followed by demethoxycurcumin and bisdemethoxycurcumin. It is possible that when ROS is produced, curcumin can scavenge its own ROS from the properties of being a relatively strong antioxidant. This leads to the higher ROS production by bisdemethoxycurcumin. Interestingly, bisdemethoxycurcumin with erythrosine could synergistically enhance the total ROS production when compared to demethoxycurcumin with erythrosine at the same concentration. We postulated that this phenomenon occurred due bisdemethoxycurcumin acting as a hydrogen bond donor to erythrosine [13], leading to enhanced ROS production by erythrosine. Further confirmation of a chemical reaction between bisdemethoxycurcumin and erythrosine should be secured for the complete utilization of this combination. 

Based on the ESR study, the type I ROS was rapidly degraded after light exposure, especially superoxide, and thus a direct comparison of the quantity of ROS produced cannot be undertaken; the only characterization of each type of ROS production was achieved by the ESR spectrum.

### 3.3. The Effect of the Methoxy Group in Curcuminoids’ Structure on ROS Production and Antifungal Activity

At the same concentration, the tested photosensitizers with bisdemethoxycurcumin as a component tended to be more effective at reducing *Candida albicans* than those with demethoxycurcumin, although not statistically significantly. This can be explained by the study of Tønnesen et al. [13]; the absence of a methoxy group in bisdemethoxycurcumin’s structure can make it a better hydrogen bond donor when compared with curcumin, and the latter is more likely to be a hydrogen bond acceptor. An effective antifungal ability relies on the ability to form a hydrogen bond between a certain substance and the fungal cell wall. Based on the good hydrogen bond donor property of bisdemethoxycurcumin, it can form a strong bond with the *C. albicans* cell wall when compared with another curcuminoid, which is followed by massive ROS formation intracellularly that can be more toxic than the extracellular ROS. On the contrary, the study of Amalraj et al. [28] mentioned that the higher the number of methoxy groups, the higher the lipophilicity of molecules in the mother nucleus, making it easier for the molecules to enter the fungal cell membrane and inhibit the growth of the fungi. Based on this hypothesis, the antifungal effect of bisdemethoxycurcumin might be lower than that of demethoxycurcumin, but in our study, a dramatic reduction in *Candida albicans* cells was observed as caused by the bisdemethoxycurcumin-containing test photosensitizer, and thus we hypothesized that the effect of ROS itself plays a more major role than the fungal cell adhesion ability of photosensitizers in *Candidiasis* suppression. This was also found in the study of Kazantzis et al. [29], showing evidence that the lack of the methoxy-aromatic substitution in bisdemethoxycurcumin favors a high degree of free radical production.

### 3.4. Influence of Nano-Titanium Dioxide on ROS Production and Antifungal Activity

In the ESR assay, the addition of 10% nano-titanium dioxide (by mass) contributes to the statistically higher formation of ROS in the erythrosine-containing group. On the other hand, in the group of single curcumin derivatives, the addition of 10% nano-titanium dioxide (by mass) did not improve the formation of ROS. According to a study by Liao et al. [30], the addition of nano-titanium dioxide to various photosensitizers can lead to peak absorption shift to the red optical absorption edge (longer wavelength). Green light is well absorbed by erythrosine, but in the present study, we used a blue light, which can also be absorbed by erythrosine. The molar attenuation coefficient (molar extinction coefficient, molar absorptivity), or the ability to absorb light at different wavelengths, of the erythrosine under green light (or at a wavelength of 532 nm) is 90,600 mol^−1^ cm^−1^ [31], but under blue light, the molar attenuation coefficient is around 20,000 mol^−1^ cm^−1^ [32]. Therefore, when the peak shift to the red region (longer wavelength), as induced by nano-titanium dioxide, occurred in the erythrosine irradiated by blue light, greater light energy absorption and ROS production were facilitated. This would not have occurred in erythrosine under green light. On the other hand, curcumin derivatives are already absorbed well in the blue light region. Upon blue light irradiation, demethoxycurcumin has a light absorption coefficient of 57,800 mol^−1^ cm^−1^, while that of bisdemethoxycurcumin is 49,500 mol^−1^ cm^−1^ [33]. All of these coefficients are almost the maximum, based on their capability, and therefore when the peak absorption is shifted to the longer wavelength by the influence of nano-titanium dioxide, the light absorption capacity does not improve; rather, the light absorption slightly decreased, and because of this a reduced level of ROS formation was evident.

Nevertheless, in the *Candida albicans* inhibition study, the addition of 10% nano-titanium dioxide did not demonstrate a statistically significantly enhanced efficacy in *C. albicans* inhibition in the erythrosine-containing groups, despite there being an increased number of ROS in the ESR assay. It was hypothesized that ROS might reach the threshold of the therapeutic dose, causing no more antifungal action. 

### 3.5. Viability of Gingival Fibroblast Cell.

To apply any photosensitizers in a clinical situation, non-toxicity must be declared as a first priority. Based on the study of Bugelski et al. [24], the acceptable cytotoxicity of certain substances is determined by cell number reductions, which should not exceed 50%. In the present study, the treatment of human gingival fibroblasts with all of the tested photosensitizers gave a fibroblast viability higher than 50% in all time periods, and thus all of the tested photosensitizers are likely to have no toxicity. Our results were consistent with those from the study of Kloesch et al., wherein free curcumin at a 20 µM concentration exposed to synovial fibroblast cells for 24 h demonstrated some degree of toxicity [34]. Nonetheless, in the present study, our samples were exposed to the original structure of all tested photosensitizers for only 15 min prior to light irradiation, which altered our photosensitizer to ROS. For this reason, we could find almost no toxicity in the normal gingival fibroblast cells. Additionally, a study of Mpountoukas et al. also demonstrated that erythrosine became toxic to DNA at 500 µM. [35]. Demethoxycurcumin and bisdemethoxycurcumin had no different levels of toxicity; however, the study of Chen et al. demonstrated an increase in antioxidant ability of photosensitizers with the methoxy group in their structure [36]. One potential limitation in the present study was that the cells were immersed directly in the test photosensitizer, but in the human body, the photosensitizer is not exposed to the cells for such a long period, due to the degradation caused by various enzymes and the flushing effect of bodily fluids in the human body, so the toxicity obtained in this experiment was greater than what it would be in reality. 

### 3.6. The Correlation of ROS Production and Candida albicans Inhibition

It was found that ROS formation and *Candida albicans* inhibition were significantly and inversely related to each other, with a low–moderate level of relationship. This phenomenon can be explained by the fact that the biological system is complicated and sophisticated, and that dead pathways involve a number of factors, thus the degree of ROS formation cannot explain *Candida albicans* inhibition. To explain more precisely *C. albicans’* growth suppression, some other factors, such as the ability of the photosensitizer to adhere to the cell walls of the fungi, the types of ROS, and the effects of reactive nitrogen species, etc., need to be considered. 

### 3.7. Limitation of the Study

In terms of its properties, erythrosine is hydrophilic, but curcumin derivatives are a hydrophobic substance that can dissolve only in ethanol and oil. Hence, both photosensitizers do not combine well. Therefore, attempts to develop a more compatible mixture are needed, such as the pre-dissolution of curcumin with ethanol, which is then lyophilized to form a powder in combination with erythrosine powder, and that will only dissolve in water.

The probe for type II ROS in ESR assay is rare [37], expensive, and the signal is not very clear, but for a complete explanation of the effect of a test photosensitizer, the measuring of type II by ESR is pivotal. Reactive nitrogen species (RNS) must also be measured because this is the other entity that might play some role in *Candida albicans* inhibition.

DMPO for the ROS measurement of the type I photodynamic reaction could not analyze erythrosine concentrations over 1000 µM, or curcumin derivatives less than 50 µM or more than 200 µM. Therefore, the experiments must use the concentration mentioned in the Materials and Methods part.

Additionally, a quantitative characterization of all the complexes obtained with nanoparticles is important in order to elucidate the adsorption kinetics of nanoTiO_2_ to photosensitizers, as well as its influence on the biological fate of *Candida albicans.* The presence of unbound nano-titanium dioxide particles may play some role different from that which our results suggest. Thus, a scanning and transmission electron microscopic study should be conducted. 

Generally, a topical substance used in the oral cavity needs to have the ability of strong bio-adhesion to oral mucous membranes, because of the flushing effect caused by saliva and gingival crevice fluid. The further development of our photosensitizer in the form of gel and orabase (wax) is recommended.

## 4. Materials and Methods

### 4.1. Photosensitizers Preparation 

Demethoxycurcumin (Sigma-Aldrich, St. Louis, MO, USA), bisdemethoxycurcumin (Sigma-Aldrich, St. Louis, MO, USA), erythrosine (Sigma-Aldrich, St. Louis, MO, USA) and nano-titanium dioxide (Sigma-Aldrich, St. Louis, MO, USA) were used as the photosensitizers. The activities of photosensitizers were tested as individual test photosensitizers or combinations, as follows:(1)Single 10 or 20 µM demethoxycurcumin and 110 or 220 µM erythrosine;(2)10 or 20 µM demethoxycurcumin with/without 110 or 220 µM erythrosine with 10% nano-titanium dioxide;(3)10 or 20 µM demethoxycurcumin with/without 110 or 220 µM erythrosine without nano-titanium dioxide;(4)10 or 20 µM bismethoxycurcumin with/without 110 or 220 µM erythrosine with 10% nano-titanium dioxide;(5)10 or 20 µM bismethoxycurcumin with/without 110 or 220 µM erythrosine without nano-titanium dioxide.

Therefore, in total, 28 photosensitizer preparations were tested.

Demethoxycurcumin and bisdemethoxycurcumin powder were dissolved in methanol and then dissolved in water to form 200 µM stock solutions. Erythrosine was prepared by dissolving erythrosine powder in distilled water to obtain a stock solution with a concentration of 2200 µM. Nano-titanium dioxide equal to 10% of the mass of curcumin derivatives and/or erythrosine was weighted and mixed with other test photosensitizers, and we adjusted the concentrations to the final concentration with water. A filtration process using a 0.22 μM filter was performed prior to use.

### 4.2. Dental light Optic Preparation

We prepared a dental light optic (VALO^®^ Ortho Cordless, South Jordan, UT, USA), with wavelength 395–480 nm and power 3200 mW/cm^2^. We mounted the light machine on a stand, placing it away from the glass slide (containing *Candida albicans* biofilm) by 2.4 cm.

### 4.3. Determination of ROS Production Using the Electron Paramagnetic Resonance (EPR) Technique 

#### 4.3.1. Total ROS Production

The total ROS production in the photodynamic reaction was estimated by decreasing the ESR signal of TEMPO (Sigma-Aldrich, St. Louis, MO, USA) [38]. The reaction mixture was composed of 90 μL of the photosensitizer preparation and 10 μL of 1 mM TEMPO (the final concentration of TEMPO was 100 μM). The mixture was irradiated with dental blue light for 21 s (with a 1 s non-irradiated session every 3 s). Then, 30 μL of each mixture was transferred into a capillary tube and placed in an ESR tube. 

The ESR signals were recorded with X-band ESR (E500, Bruker, Billerica, Lowell, MA, USA) equipped with an ELEXSYS super-high-sensitivity probehead cavity. The parameters of the electron paramagnetic resonance spectrometer were as follows: center field 3510 Gauss, modulation frequency 100 kHz, modulation amplitude 1 Gauss, power 2 mW, receiver gain 80 dB, sweep time 41.94 s, constant time 1.28 m. The ESR spectrum was recorded at 3 and 5 m after light exposure. To quantify the concentration of ROS, the diminishing area under peak of TEMPO was measured in comparison with 100 μM TEMPO. Measurements were conducted in triplicate.

#### 4.3.2. ROS Measurement of Type I Photodynamic Reaction

5,5-Dimethyl-1-pyrroline-*N*-oxide (DMPO) (Enzo Life Science, Farmingdale, South Farmingdale, NY, USA) was used as a spin trapping agent for the quantification of hydroxyl radical. The reaction mixture was composed of 10 μL test photosensitizer preparation, 80 μL of water and 10 μL of DMPO 10 μM, with a final concentration of DMPO of 100 μM. Then, the mixture was irradiated with dental blue light and we performed the ESR measurement as described above. The ESR signal intensities of the DMPO-OH and DMPO-OOH adducts were measured to estimate the production of hydroxyl radicals and superoxide radicals, respectively. 

### 4.4. Toxicity Testing of A Test Photosensitizer Containing Curcumin Derivatives, Erythrosine, and Nano-Titanium Dioxide to Gingival Fibroblast Cell by PrestoBlue^®^

#### 4.4.1. Test Photosensitizers Used in the Experimental Group

All test photosensitizers containing 220 µM erythrosine were mixed with 10 or 20 µM curcumin derivatives with/without 10% *w*/*w* nano-titanium dioxide, and a total of 8 test photosensitizers were subjected to the viability assay. We diluted all test photosensitizers with basal medium (DMEM, Antibiotic-Antimycotic™ 1%, GlutaMAX™ 1%). The gingival fibroblast cells cultured with hydrogen peroxide were set as the positive control, and DMEM + 10% *v/v* fetal bovine serum was set as the negative control.

#### 4.4.2. Gingival Fibroblast Cell Culture Conditions

We cultured the gingival fibroblast cell lines in a t25 culture flask using a mixed medium of DMEM (Dulbecco’s Modified Eagles, Gibco, Waltham, MA, USA), 10% (*v/v*) fetal bovine serum (Invitrogen, Waltham, MA, USA.), Antibiotic-Antimycotic™ (Amphotericin B, Penicillin, Streptomycin) 1%, and GlutaMAX™ 1% (Invitrogen, Waltham, MA, USA). We then incubated this in an incubator (Shel Lab, Sheldon, Cornelius, OR, USA) that controls volume, with the following parameters: carbon dioxide 5%, temperature 37 °C, humidity 95%. We changed the cell culture every 2–3 days until the cells grew to 90% confluence.

#### 4.4.3. Study Process

We diluted the cell culture to obtain 10^5^ cells/mL. We then put 100 μL of medium with cells into each of the 96-well plates, to get a total of 10^4^ cells/well. We incubates this in an incubator set up with the above-mentioned conditions for 24 h. We changed the complete medium to serum-free medium and incubated for 45 min. Then, we removed the serum-free medium and loaded the test photosensitizer mixed with the new serum-free medium for 15 min. This was irradiated with blue dental LED light for 27 s (3 s of irradiation alternately with 1 s between each session). We then measured the cell viability at 6 and 24 h, at the specified time, and discarded the substance out of the well. We filled up each of the 96-well plates with 10 μL of PrestoBlue™ in serum-free medium, and then incubated these and measured the fluorescent intensity at excitation wavelength 560 nm and emission wavelength 590 at 90 and 120 min. Then, we calculated the viability of the cells using the formula recommended by the manufacturer.

### 4.5. Candida albicans Inhibition Assay

#### 4.5.1. Test Photosensitizers Used in the Experimental Group

There were 22 test photosensitizers from the ESR assay in the experimental group to test for *Candida albicans* inhibition. Nystatin (1:100,000 U/mL) oral suspension was set as the positive control and phosphate buffer saline as the negative control.

#### 4.5.2. *Candida albicans* Biofilm Culture 

*Candida albicans* ATCC 10231 was grown on Sabouraud dextrose broth, then blended in a high-speed blender at 350× *g* for 10 min, and washed with phosphate buffer saline twice. We then adjusted the density with a spectrophotometer at the wavelength 530 nm at 0.381. The solution containing yeast *Candida albicans* in the form of 10^7^ cells/mL final concentration was achieved.

Biofilms were produced in 6-well plates based on but modified from Jin et al. [38] and Thein et al. [39], with glass coverslips, and incubated for 1.5 h at 37 °C in an orbital shaker at 75 rpm to promote yeast adherence to the surfaces of the wells (adhesion phase). Then, we added 1× yeast nitrogen base and 50 mM glucose for promoting biofilm growth, and then continued incubating for 48 h to acquire mature *Candida* biofilm.

#### 4.5.3. Study Process

We took the glass slide containing the *Candida* biofilm to the study plate, and added 2 mL of test photosensitizer onto the glass containing the *Candida albicans* biofilm. We irradiated with the blue dental LED light for 27 s (3 s of irradiation alternately, with 1 s between each session). Upon irradiation completion, this was vibrated with an ultrasonic incubator at 75 rpm for 15 min to separate the unbound colonies. We cultivated a *Candida albicans* suspension in Sabouraud dextrose agar at 37 °C for 48 h. Then, we counted the colonies of *Candida albicans* (CFU/mL) by the drop plate technique at 1:1000 dilutions, and a logarithm transformation of CFU/mL was performed.

### 4.6. Statistical Analysis

All experiments were performed in triplicate (the analysis of *Candida albicans* in CFU between groups, numbers of living cells, and ROS formation measured by descriptive statistics, such as means and standard deviation). We measured the distribution with the Shapiro–Wilk test. Inferential statistics were used to analyze the differences in the numbers of *Candida albicans* in log_10_ CFU/mL between groups and the differences in the numbers of living cells between groups at 6 and 24 h by using the Kruskal–Wallis and Dunn test. Additionally, we used one-way ANOVA with post hoc test to determine ROS formation. 

## 5. Conclusions

In conclusion, it was found that 20 μM bisdemethoxycurcumin + erythrosine 110–220 μM + 10% nano-titanium dioxide (by mass) tended to generate relatively high ROS, and effectively inhibited *Candida albicans* without inducing cytotoxicity to normal human gingival fibroblast cells.

## 6. Patents

The test photosensitizer used as a combination photosensitizer in the present study was approved by patent (Filed Number 2003000412).

## Figures and Tables

**Figure 1 molecules-26-02405-f001:**
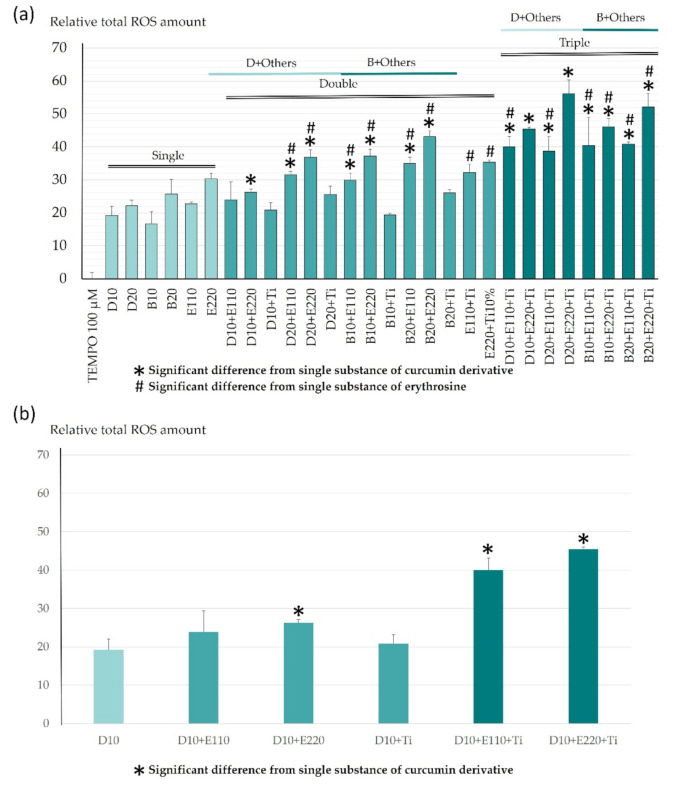
Total ROS formation of the test photosensitizer preparations used in photodynamic therapies with dental blue light optics (72 J/cm^2^) using ESR spectroscopy and TEMPO as a spin probe. (**a**) All test photosensitizers; (**b**) 10 µM demethoxycurcumin and its combination; (**c**) 20 µM demethoxycurcumin and its combination; (**d**) 10 µM bisdemethoxycurcumin and its combinations; (**e**) 20 µM bisdemethoxycurcumin and its combinations.

**Figure 2 molecules-26-02405-f002:**
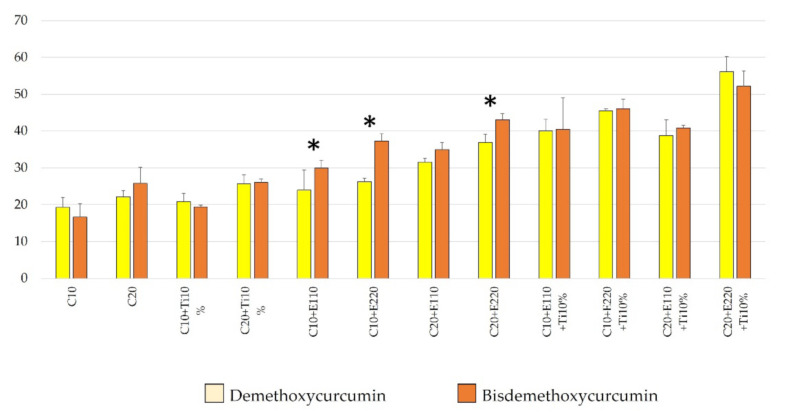
Comparison of ROS derived from the test photosensitizer between curcumin derivatives at the same concentrations using ESR. (*n* = 3) C = concentration, yellow column = demethoxycurcumin and orange column = bisdemethoxycurcumin. * = significant difference at (*p* < 0.05) between groups with the same component.

**Figure 3 molecules-26-02405-f003:**
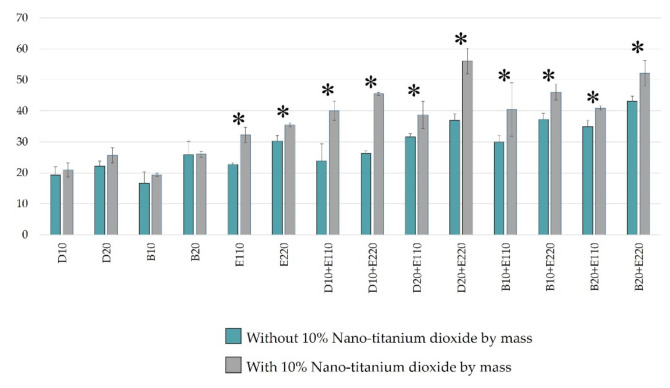
Comparison of ROS between test photosensitizers with and without 10% nano-titanium dioxide by mass using ESR (*n* = 3). Green column = without 10% nano-titanium dioxide and grey column = with 10% nano-titanium dioxide. * = significant difference at (*p* < 0.05) between groups with the same component.

**Figure 4 molecules-26-02405-f004:**
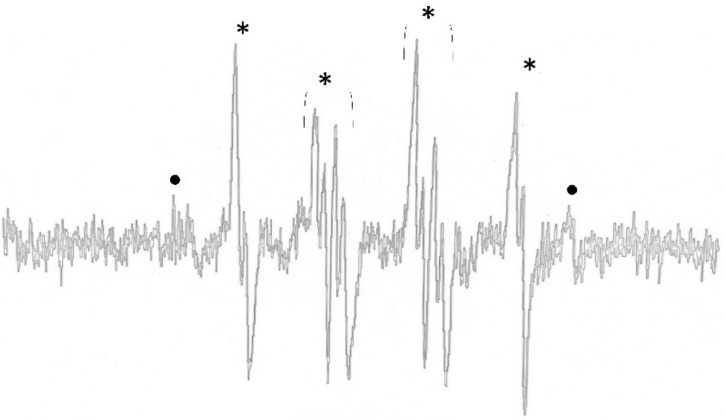
Representative spectrum of DMPO adduct 30 s after blue light irradiation of photosensitizers. The DMPO-OH adduct is indicated by ● and the DMPO-OOH adduct is indicated by *.

**Figure 5 molecules-26-02405-f005:**
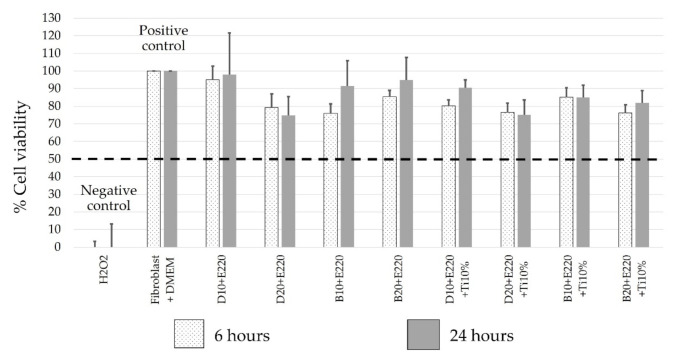
Comparison of percentage of human gingival fibroblast viability among 8 test photosensitizers, which includes 10/20 µM curcumin derivatives, and 220 µM erythrosine with/without 10% nano-titanium dioxide (by mass) in photodynamic therapies after 6 and 24 h, via PrestoBlue^®^ vitality assay. DMEM and H_2_O_2_ were used as the positive and negative control, respectively. Kruskal–Wallis with Dunn post hoc tests were performed to observe the significant differences among the experimental groups (*p* < 0.05), (*n* = 9). The dashed line shows the level of 50% viability, proposed by Bugelski et al. [24].

**Figure 6 molecules-26-02405-f006:**
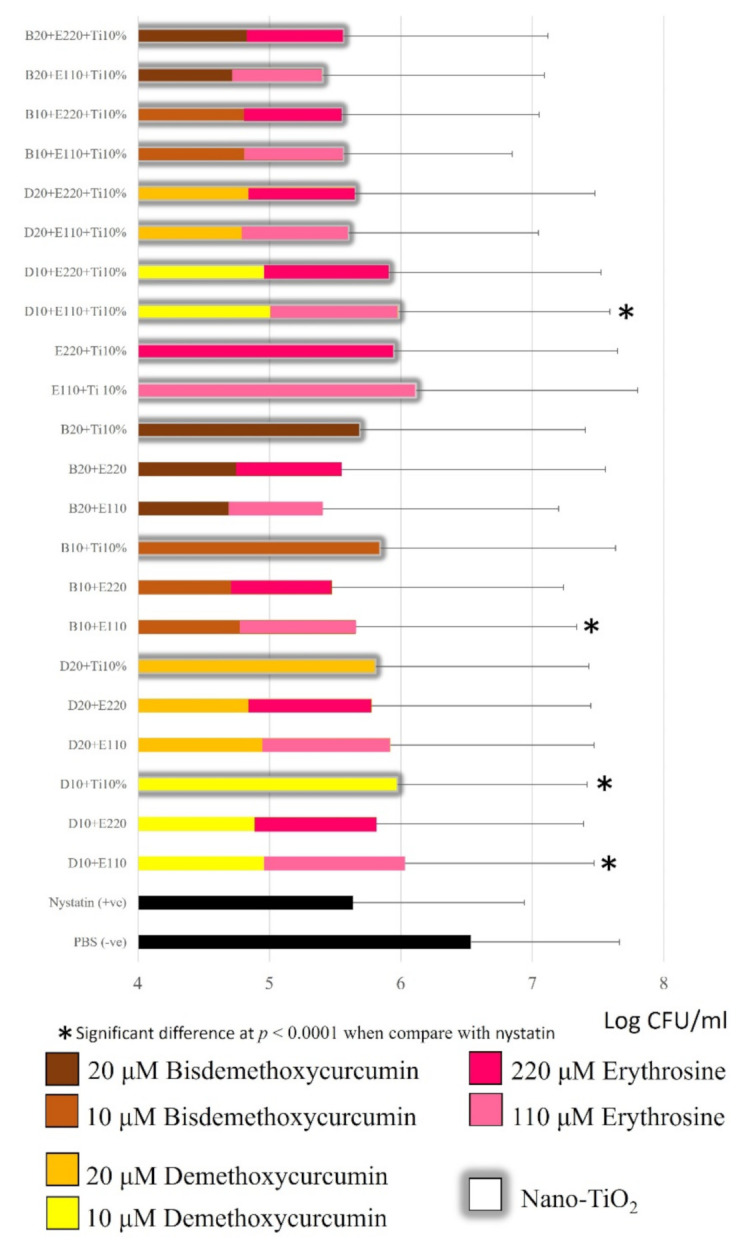
Comparison of *Candida albicans* numbers in log_10_ CFU after treatment with test photosensitizers at 48 h using a drop plate assay. All photosensitizers were incubated for 15 min prior to blue light (395–480 nm) irradiation 72 J/cm^2^. Phosphate buffer saline was set as the negative control and nystatin oral suspension (1:100,000 units) was set as the positive control. Kruskal–Wallis with Dunn post hoc tests were performed to assess the significant difference when compared among experimental groups (*n* = 9). * Significant difference at *p* < 0.0001 when compared with nystatin.

**Figure 7 molecules-26-02405-f007:**
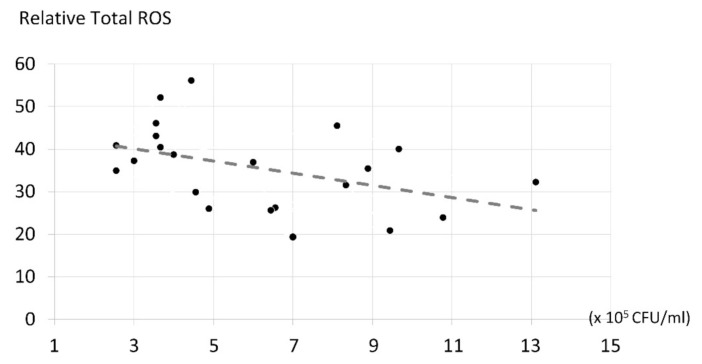
The correlation coefficient of ROS and *Candida albicans* inhibition of the experiments testing photosensitizers. Spearman’s rank correlation coefficient testing was performed.

## Data Availability

Data are contained within this article.

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
