# Peer review of "Inhibitory Effects of Erythrosine/Curcumin Derivatives/Nano-Titanium Dioxide-Mediated Photodynamic Therapy on Candida albicans"

_molecules, 2021, doi:10.3390/molecules26092405_

Round 1

Reviewer 1 Report

The English must be improved. Generally, the word order is not adequate, creating confusion of the sense.

The binomial nomenclature is always written in italics. Also, the name of the Genus is capitalized.

The reference Bugelski et al. it is not numbered in the text nor is it included in the reference list; this becomes [24] and all subsequent quoted must be renumbered.

Supplementary suggestions are included in the manuscript, as comments.

Author Response

The English must be improved. Generally, the word order is not adequate, creating confusion of the sense.

 Adrress  We have checked and correct accordingly. And this manuscript was checked by certified Language expert in our institute.

The binomial nomenclature is always written in italics. Also, the name of the Genus is capitalized.

Address  Thank you for your kind advice, I have used the navigator function and substituted all candida to Candida.

The reference Bugelski et al. it is not numbered in the text nor is it included in the reference list; this becomes [24] and all subsequent quoted must be renumbered.

Address   We are sorry for this mistake. We have added the reference from Bugelski et al and renumber all references accordingly totally 41 references.

Supplementary suggestions are included in the manuscript, as comments.

Address  We carefully corrected all suggested points.

Line 4 change albican to albicans.

Line 27-28 Changed from “the toxicity test to gingival fibroblast cells at 6 and 24 hours was performed using Presto blue® on the 8 highest ROS generated test

photosensitizers…”  to “the PrestoBlue® toxicity test of the gingival fibroblast cells was performed at 6 and 24 hours on 8 highest ROS generated test photosensitizers…”

Line 31    Added 1 space between 48 and hrs

Line 55    Added comma after “albicans,”

Line 64    Added the abbreviation of photodynamic therapy “(PDT)” therefore the word

PDT will be used in the later sessions of the manuscript.

Line 122-3  The word “more antimicroorganixms than” was added because this is the

comparative sentence as suggested. 

Line 136   We changed to position of the 1st anatase to “(anatase form)” and moved to

after the word “dioxide”

Line 137    We changed the word “of” to “on” as suggested.

Line 222     We corrected the word prediations” to “conditions”.

Line 235     We added the information about hydrogen peroxide (50 w/w) already.

Line 247     We added [24] after the word Bugelski et al.

Line 292      We corrected the word “irritated” to “irradiated”

Line 301      We inserted comma after the word “density” as suggested.

Line 351      We corrected to name of the author from “Augustine” to “Amalraj”

already.

Line 378-380   We rearranged the sentence from “longer wavelength by the influence of nano titanium dioxide, the light absorption did not improve rather slightl decrease in the formation of ROS was evident.”  To

             “longer wavelength by the influence of nano titanium dioxide, the light absorption did not improve rather slightly decreased in light absorption, because of this decreased formation of ROS was evident.”   

Line 390-391  Changed  “the acceptable cytotoxicity of certain substances determined by cell numbers should not be less than 50%” to

“the acceptable cytotoxicity of certain substances is determined by cell numbers reductions that should not exceed 50%”

Line 399-400  We adjusted the sentence from “And the study of Mpountoukas et al erythrosine became toxic to DNA at 500 µM. ” to

“And a study of Mpountoukas et al also demonstrates that erythrosine became toxic to DNA at 500 µM.”

Line 402     We added “et al” after “Chen”

Line 469      We change the content in parenthesis from “ESR” to “EPR” because our assay was electron paramagnetic resonance.

Reviewer 2 Report

Photodynamic therapy has been actively developing in recent years. The search for new multicomponent photosensitizers and protocols for their use remains an urgent task. The manuscript may be of interest to many readers, but it requires minor revisions and additions.

Isolation and purification of the complexes obtained are not described in the manuscript. Unfortunately, the quantitative characterization of the complexes obtained with nanoparticles is also omitted. The adsorption of added components on the surface of nanoparticles is rarely quantitative. It is possible that unbound components may be present in the complexes investigated. This does not allow us to draw unambiguous conclusions.

After additional editing of the text by the authors, the manuscript can be accepted for publication. Some correction.

519 solution containing yeast Candida Albicans in the form of 107 cells/ml. final concentration.

Have to be

519 solution containing yeast Candida albicans in the form of 107 cells/ml final concentration.

Author Response

Photodynamic therapy has been actively developing in recent years. The search for new multicomponent photosensitizers and protocols for their use remains an urgent task. The manuscript may be of interest to many readers, but it requires minor revisions and additions.

Isolation and purification of the complexes obtained are not described in the manuscript. Unfortunately, the quantitative characterization of the complexes obtained with nanoparticles is also omitted. The adsorption of added components on the surface of nanoparticles is rarely quantitative. It is possible that unbound components may be present in the complexes investigated. This does not allow us to draw unambiguous conclusions.

Address  Thank you for your kind advice.

         First of all, we used the cell culture grade of all substances as indicated at the Line 435-437, therefore we did not perform isolation. Purifications by filtrations were performed using 0.22 μM filtration syringe. We added this in Line 463-464

Also, because the quantitative characterization of the complexes is not performed, we have included this point in the last part of Discussion topic 3.7 Line 432-437.

After additional editing of the text by the authors, the manuscript can be accepted for publication. Some correction.

519 solution containing yeast Candida Albicans in the form of 107 cells/ml. final concentration.

Have to be

519 solution containing yeast Candida albicans in the form of 107 cells/ml final concentration

Address   Thank you for your kind advice. We have corrected it as shown in Line 532